# Physicochemical Quality Retention during Cold Storage of Prepackaged Barramundi Meat Processed with a New Microwave-Assisted Induction Heating Technology

**DOI:** 10.3390/foods12163140

**Published:** 2023-08-21

**Authors:** Chiu-Chu Hwang, Hung-I Chien, Yi-Chen Lee, Jun-Cheng Kao, Yu-Ru Huang, Ya-Ling Huang, Chun-Yung Huang, Yung-Hsiang Tsai

**Affiliations:** 1Department of Seafood Science, National Kaohsiung University of Science and Technology, Kaohsiung 811213, Taiwan; omics1@gmail.com (C.-C.H.); lionlee@nkust.edu.tw (Y.-C.L.); f109176108@nkust.edu.tw (J.-C.K.); ylhuang@nkust.edu.tw (Y.-L.H.); cyhuang@nkust.edu.tw (C.-Y.H.); 2Department of Food Science, National Ilan University, Ilan 260, Taiwan; yrhuang@ems.niu.edu.tw

**Keywords:** microwave, microwave-assisted induction heating, quality retention, storage life, barramundi

## Abstract

Microwave-assisted induction heating (MAIH) is a composite microwave and induction heating to supply rapid and uniform heating of food. A recent study showed that the optimum MAIH heating condition for barramundi meat was 90 °C/110 s or 70 °C/130 s. This study examines whether the microwave-assisted induction heating (MAIH) technology (at 70 °C for 130 s or 90 °C for 110 s) can more effectively slow down the quality loss of barramundi meat during cold storage than the traditional boiling method (at 90 °C for 150 s). The results show that no microbial growth was observed in the three groups of heated barramundi meat samples during the 60 days of cold storage. However, the MAIH technology slowed down the increase in the total volatile basic nitrogen (TVBN) content more significantly than the boiling method. As the cold storage time increased, though, the *L** (lightness), *a** (redness), and *W* (whiteness) values decreased, while the *b** (yellowness) and color difference (Δ*E*) values increased in the three treatment groups. However, the MAIH technology slowed down the decrease in the *L**, *a**, and *W* values more significantly, and produced a Δ*E* value smaller than the boiling method. Moreover, the MAIH technology ensured higher hardness and chewiness of the barramundi meat than the boiling method. Overall, the MAIH technology slowed down the quality loss of the barramundi meat and maintained better color and texture during cold storage.

## 1. Introduction

Microwaves are electromagnetic waves with a frequency range of 300 MHz to 300 GHz. According to the Federal Communications Commission (FCC), the working frequency of household microwave ovens is 2.45 GHz. The working frequency of industrial microwave ovens is mostly 915 MHz [1]. Microwaves have better penetrability because of their longer wavelength compared with other types of electromagnetic waves (e.g., infrared waves and far infrared waves) [2,3]. Despite various advantages (e.g., rapid heating and high efficiency), microwave heating has the disadvantage of uneven heating, resulting in cold and hot spots in the heated food [4]. Thus, microwave heating may fail to achieve the ideal sterilization effect at the cold spots and overheat or burn food at the hot spots [5].

As a hybrid or combination of microwave and traditional processing technologies, microwave-assisted processing technology has received increasing attention in recent years. By integrating the advantages of both microwave and traditional processing technologies, this technology can save energy, improve processing quality, and reduce processing time and manufacturing costs [1,6]. Microwave-assisted processing technology has many applications, including microwave-assisted freeze drying (MAFD), microwave-assisted vacuum frying (MAVF), microwave-assisted oven-frying (MAOF), and microwave-assisted infrared heating (MAIRH) [5,7,8].

Microwave-assisted induction heating (MAIH) technology is a modular composite microwave heating system with a separable cavity (Figure 1A). The upper half-cavity is a microwave heating unit with a microwave feed-in device, and the lower-half cavity is an electromagnetic induction heating unit with a conventional heat source. The two parts are combined to form a microwave resonant cavity (Figure 1B) [9]. MAIH technology has numerous advantages, including higher heating temperature, higher heating efficiency, the ability to operate at atmospheric pressure, and uniform heating due to cavity rotation [10].

White shrimps heated using MAIH technology (at 130 °C/80 s or 90 °C/100 s) were cooked thoroughly without detectable microbes and could maintain a minimum cooking loss [9]. A comparison between the MAIH technology under the above conditions with microwave (MW) or induction heating (IH) alone showed that the shrimps processed by MW or IH method were not fully cooked, and cold spots and uneven temperature distribution were observed in their thermal images, whereas the shrimps processed by MAIH were fully cooked, and even temperature distribution was observed in their thermal images [9]. Moreover, compared with the conventional boiling method, the MAIH technology could extend the storage life of white shrimps by slowing down microbial growth and freshness loss [9]. It also facilitated effective shell removal and pasteurization when used to heat hard clam samples under appropriate conditions (at 130 °C/110 s or 90 °C/130 s) [11]. In fact, the MAIH technology could slow down microbial growth and the decline in the physicochemical quality of clam samples during their storage life [12]. Thus, MAIH is an effective emerging technology for heat processing of food.

A recent study has shown that if barramundi meat was heated using the MAIH technology under optimal conditions (at 90 °C/110 s or 70 °C/130 s), it can be evenly heated and fully cooked without detectable microbes [13]. However, no studies have been conducted to examine whether the MAIH technology can improve the microbial and physicochemical quality of fish meat in low-temperature storage. This study investigates whether the MAIH technology can heat barramundi meat to slow down its quality loss during its cold storage life better than the conventional boiling method. To this end, barramundi meats heated with the MAIH technology under optimal conditions (at 90 °C for 110 s and 70 °C for 130 s), as well as the conventional boiling method (at 90 °C for 150 s), were stored in refrigerator, and the changes in the bacterial count and the physical and chemical quality of the meats were regularly analyzed.

## 2. Materials and Methods

### 2.1. Preparing Barramundi Meat Samples

The fresh barramundi (*Lates calcarifer*) specimens purchased from a fish market were put in crushed ice and transported to our lab. In the laboratory, two slices of meat were taken from the dorsal part of the fish, skinned, and cut into smaller pieces (size: 3 × 3 × 1.5 cm; weight: 20 g). Five pieces of fish (total weight: 100 g) were selected and put in a crystallized polyethylene terephthalate (CPET) box (13 cm in diameter, 3.0 cm in height, and 0.25 cm in thickness), and 150 mL 0.9% NaCl water was added to the CPET container. Finally, the CPET box was sealed with polyethylene terephthalate (PET) film (80 µm in thickness) for subsequent heating with the MAIH technology. All CPET boxes and PET films were supported by Bottle Top Machinery Co., Ltd., Taiwan.

### 2.2. Conditions for Heating Barramundi Meat with MAIH and Boiling

Before heating, the sealed CPET box with the barramundi meat samples was set in the induction half-cavity. Thereafter, the induction half-cavity was lidded and locked, and placed at the correct place above the induction heating unit (Figure 1B). The MAIH equipment (Bottle-Top Aligo-S^TM^, Nantou, Taiwan) was set with a power of 1000 W and frequency of 2450 MHz for the microwave heating unit, and two heating temperature values (90 °C and 70 °C) were set for the induction heating unit (1500 W). The total heating time was set to 110 s at 90 °C (MAIH 90) and 130 s at 70 °C (MAIH 70) [13]. After the heating process, the induction half-cavity was put in crushed ice and cooled for 7 min, and the CPET sealed box was removed for storage testing. For the traditional boiling method, the heating pot containing 0.9% NaCl water was heated to 90 °C. Thereafter, the pieces of fish were put into the heating pot and heated for 150 s. Subsequently, the pieces were taken out of the heating pot, air-dried, and quickly put into a 3M™ aseptic bag (polyethylene, 63.5 µm in thickness). Finally, the bag was cooled in crushed ice for storage testing.

### 2.3. Storage Test

The barramundi fish pieces heated with the MAIH technology under the optimal conditions (at 90 °C for 110 s and 70 °C for 130 s) and those heated by the conventional boiling method were respectively stored at 4 °C for 60 days. Each one of the group experiments was conducted in three independent boxes or bags for each storage time. Every ten days, the samples were analyzed for microbial count, colors (*L**, *a**, *b**, *W*, and Δ*E*), texture, total volatile basic nitrogen (TVBN) content, and pH value. Within the same sampling time, fish meat samples contained in the three separate boxes or bags (three replicates) were taken for each heating condition group to conduct the following microbial and physicochemical quality analyses.

### 2.4. Testing the Microbial Quality

The microbial quality test items included aerobic plate count (APC), psychotropic bacteria count (PBC), H_2_S-producing bacteria count (HBC), *Vibrio parahaemolyticus*, coliform, and *Escherichia coli*. The spread plate method was used for APC analysis; specifically, 10 g fish meat was evenly mixed with 90 mL sterile saline water, and a series of 10-fold continuous dilutions was conducted on the homogenate using sterile saline water. Homogenate worth 0.1 mL and 10-fold diluent (two replicates) were applied to the trypticase soy agar (TSA) (Difco, BD, Sparks, MD, USA), and the TSA was kept in an incubator at 30 °C for 24 to 48 h. After bacterial culture, the colony count (expressed as log CFU/g) on the TSA plate was measured [13]. To analyze the PBC, 0.1 mL of the aforesaid fish homogenate and 10-fold diluent (two replicates) were applied to the TSA, and the TSA was kept for 10 days at 7 °C in an incubator. After bacterial culture, the colony count (expressed as log CFU/g) on the TSA plate was measured [13]. To analyze the HBC, 0.1 mL of the aforesaid fish homogenate and 10-fold diluent (two replicates) were applied to the triple sugar iron agar (TSI) substrate, which was kept for 5 days at 20 °C in an incubator; thereafter, the HBC on the TSI plate was measured [13]. *V. parahaemolyticus* was measured using the CHROMagar™ Vibrio (CHROMagar, Paris, France). Coliform and *E. coli* were analyzed using the 3M Petrifilm E. coli/Coliform Count Plate (3M Microbiology, St. Paul, MN, USA). The APC, PBC, HBC, *V. parahaemolyticus,* coliform, and *E. coli* measurements of each sample were performed in duplicate.

### 2.5. Numeric Analysis of Colors

Color parameters (*L**: lightness; *a**: +redness to −greenness; *b**: +yellowness to −blueness) were measured using a colormeter (CR-300 Chroma meter, Konica Minolta, Inc., Tokyo, Japan) with 30 mm aperture opening size, D/8 optical geometry, 10° viewing angle, and illumination of D_65_, to detect color changes in the barramundi meat during cold storage. Each fish piece was measured thrice from different positions. The color measurement of each sample was performed in duplicate. The W and Δ*E* values were calculated respectively using Equations (1) and (2) [13]. The W and Δ*E* values were calculated based on the *L**, *a**, and *b** values:(1)$$W=100-\sqrt{{\left(100-L{}^{*}\right)}^{2}+a{{}^{*}}^{2}+b{{}^{*}}^{2}}$$
(2)$$\Delta E=\sqrt{{\left(L{}^{*}-L_{0}{}^{*}\right)}^{2}+{\left(a{}^{*}-a_{0}{}^{*}\right)}^{2}+{\left(b{}^{*}-b_{0}{}^{*}\right)}^{{}_{2}}}$$

Note: *L*_0_*, *a*_0_*, and *b*_0_* denote the color values of the fish meat on the 0-th day; *L**, *a**, and *b** denote the color values of the fish meat during cold storage.

### 2.6. Texture Analysis

In this study, texture profile analysis (TPA) was conducted on the barramundi meat during cold storage using a TA.XT2 texture analyzer (Stable Micro System Ltd., Surrey, UK). The TPA covered hardness, cohesiveness, springiness, and chewiness. The fish meat was put on a support pedestal with the probe perpendicular to the sample. Each fish piece was measured from three different positions. A two-cycle compression test was performed using a TA18 spherical probe (diameter: 12.7 mm,) with a target value of 3.00 mm, a pre-test speed of 1 mm/s, a test speed and return speed of 1.5 mm/s, and a trigger point load of 0.1 g [13]. The texture measurement of each sample was performed in duplicate.

### 2.7. TVBN Content and pH Value

Fish meat of 5 g was homogenized with 20 mL 6% trichloroacetic acid (TCA) using a homogenizer (IKA T18 basic, Ultra-Turrax, IKA-Werke GmbH & Co. KG, Staufen, Germany) (6000 rpm for 60 s), and thereafter, the homogenate was centrifugally filtered. After the precipitate was taken to repeat the above step, the two filtrates were collected and added with 6% TCA to the constant volume of 50 mL using 6% TCA. The TVBN content of the fish meat was measured using Conway’s dish microdiffusion method [14]. Specifically, 1 mL filtrate and 1 mL saturated K_2_CO_3_ solution were applied to the outer ring of the Conway’s dish, 1 mL boric acid solution was applied to its inner ring, and finally, the Conway’s dish was lidded and kept for 90 min at 37 °C in an incubator. After the reaction was completed, the boric acid solution was titrated using 0.02-N HCl solution and the TVBN value was expressed in mg/100 g of fish meat. The TVBN measurement of each sample was performed in duplicate. Additionally, a 10 g fish meat sample was homogenized with 90 mL deionized water for 90 s using a homogenizer (IKA-T25 Ultra-Turrax, IKA-Werke GmbH & Co, Staufen, Germany). After the fish meat sample was homogenized, the pH value of the homogenate was detected using a pH meter (HI1230, Hanna Instruments, Smithfield, RI, USA) [13]. The pH measurement of each sample was performed in duplicate.

### 2.8. Statistical Analysis

In this study, all values were derived as the average ± standard deviation of three individual containers or bags and two-way analysis of variance (ANOVA) was conducted using the SPSS (SPSS for Windows, version 21, IBM Corp., Armonk, NY, USA) to analyze the effect of three types of treatments (MAIH 70 °C, MAIH 90 °C, and boiled) and the treatment time on different measurements. Tukey’s HSD test was used to determine the differences in the data measured under different heating conditions and different storage periods. When the *p* value was smaller than 0.05, data were considered statistically different.

## 3. Results and Discussion

### 3.1. Changes in the Microbial Count of the Barramundi Meat Heated with Different Methods during Cold Storage

Table 1 describes the microbial test results of the barramundi meat that was heated using different methods (i.e., boiling at 90 °C, MAIH at 70 °C, and MAIH at 90 °C) and stored at 4 °C for 60 days. For raw barramundi meat, the APC, PBC, HBC, *V. parahaemolyticus*, coliform, and *E. coli*, counts were approximately 3.85, 3.56, 3.20, <2.0, 2.10, and <1.0 log CFU/g, respectively. During the 60 days of cold storage, no APC, PBC count, *V. parahaemolyticus*, coliform, or *E. coli* were detected in the three treatment groups (boiling, MAIH 70 °C, and MAIH 90 °C). Evidently, the MAIH technology (at 90 °C for 110 s and 70 °C for 130 s) could achieve the same fresh-keeping effect as boiling (90 °C for 150 s); moreover, the MAIH technology shortened the heating time. The APC and PBC of the boiled white shrimps continued to increase during cold storage, whereas the APC and PBC in MAIH-heated white shrimps were not detected until the 16th day of cold storage [15]. Likewise, the APC and PBC of boiled hard clams continued to increase during cold storage, whereas the APC and PBC of MAIH-heated hard clams were not detected until the 20th day of cold storage [12]. Evidently, the traditional boiling method cannot achieve complete pasteurization, whereas the MAIH technology can effectively slow down the microbial growth of white shrimp and hard clams during cold storage [12,15]. However, this study found that both boiling and MAIH could kill microorganisms in the barramundi meat and no microbial growth was observed in the meat during cold storage. This may be due to the fact that the microbial count of the fresh barramundi meat (<4.0 log CFU/g) was significantly smaller than that of the white shrimps (6.24 log CFU/g) and the hard clams (5.40 log CFU/g), and the microbes in the fresh barramundi meat were easily killed by heating [9,11].

### 3.2. Changes in the Color Values of the Barramundi Meat Heated with Different Methods during Cold Storage

Table 2 describes the changes in the color values of the barramundi meat that was heated using different methods (traditional boiling, MAIH at 70 °C, and MAIH at 90 °C) and stored at 4 °C for 60 days. The *L**, *a**, and *W* values decreased with the increasing storage time in the three treatment groups (boiling, MAIH 70 °C, and MAIH 90 °C); within the same storage time, the *L**, *a**, and *W* values in the two treatment groups of MAIH were higher than those in the treatment group of boiling (*p* < 0.05). Conversely, the *b** and Δ*E* values increased with the increasing storage time; within the same storage time, the Δ*E* value in the treatment group of boiling was higher than the Δ*E* values in the two treatment groups of MAIH (*p* < 0.05). At the end of cold storage (on the 60th day), the *L** (79.18), *a** (0.95), and *W* (75.36) values in the treatment group of MAIH 70 °C and *L** (78.31), *a** (0.93), and *W* (75.05) values in the treatment group of MAIH 90 °C were higher than those in the treatment group of boiling (*p* < 0.05). Conversely, the Δ*E* (6.98) value in the treatment group of boiling was higher than the Δ*E* values in the two treatment groups of MAIH (*p* < 0.05) (Table 2). The results of this study are similar to those of the study conducted by Li, Bland, and Bechtel [16], which found that the *L** value (lightness) of microwave-heated catfish meat was higher than that of boiled catfish meat. Likewise, the MAIH technology significantly slowed down the color fading of white shrimps during cold storage compared with the conventional boiling method [15]. Thus, in the three treatment groups, the lightness (*L**), redness (*a**), and whiteness (*W*) of the barramundi meat decreased with the increasing cold storage time, whereas the yellowness (*b**) and Δ*E* of the barramundi meat increased with the increasing cold storage time. This indicates a gradual fading of the colors and the brightness of the appearance of the barramundi meat, which may be because the browning substance generated by the Maillard reaction between reducing sugar and amino acid in the barramundi meat increased with the increasing cold storage time, resulting in a yellowish-brown color [17]. Moreover, the MAIH technology could slow down the color fading of the barramundi meat throughout 4 °C storage more effectively than the traditional boiling method.

### 3.3. Changes in the Texture of the Barramundi Meat Heated Using Different Methods during Cold Storage

Figure 2 shows the changes in the texture (hardness, cohesiveness, springiness, and chewiness) of the barramundi meat that was heated using different methods (traditional boiling, MAIH at 70 °C, and MAIH at 90 °C) and stored at 4 °C for 60 days. As shown in Figure 2A, the initial hardness values of the barramundi meat in the three treatment groups (boiling, MAIH 70 °C, and MAIH 90 °C) were 87.38 (g), 120.00 (g), and 122.58 (g), respectively; the hardness of the barramundi meat slowly increased with the increasing cold storage time in the three treatment groups; during the entire cold storage life, the hardness of the barramundi meat in the two treatment groups of MAIH was significantly higher than that in the treatment group of boiling (*p* < 0.05). Within the same cold storage time, the hardness of the barramundi meat did not significantly differ between the MAIH 70 °C and the MAIH 90 °C treatment groups (*p* > 0.05). The hardness of microwave-heated salmon slices was higher than that of boiled salmon slices [17]. Additionally, the MAIH technology significantly slowed the decline in the hardness of white shrimps during cold storage compared with the traditional boiling method [15]. During cold storage at 4 °C, the hardness of MAIH-heated hard clam meat was higher than that of boiled hard clam meat [12]. Overall, the hardness of the MAIH-heated barramundi meat was higher than that of the boiled barramundi during cold storage. This may be because microwave heating denatured the protein in the fish meat and reduced its water content, resulting in the contraction and compacting of its muscle tissues, and the consequent higher hardness [17].

As shown in Figure 2B,C, cohesiveness and springiness did not vary significantly during the entire cold storage life, and their values were not significantly different between the three treatment groups (*p* > 0.05). Chewiness shared a similar variation trend with hardness (Figure 2D). The chewiness values in the three treatment groups (boiling, MAIH 70 °C, and MAIH 90 °C) were initially 1.62 (mJ), 2.12 (mJ), and 2.21 (mJ), respectively, and increased with the increasing cold storage time. During the same cold storage time, the chewiness in the two treatment groups of MAIH was higher than that in the treatment group of boiling (*p* < 0.05). The MAIH technology significantly slowed the decline in the chewiness of white shrimp during cold storage compared with the traditional boiling method [15]. During cold storage at 4 °C, the chewiness of MAIH-heated hard clam meat was higher than that of boiled hard clam meat [12]. Thus, the MAIH-heated barramundi meat had higher hardness and chewiness than the boiled barramundi meat during cold storage, proving that the MAIH-heated barramundi meat had good texture during cold storage. Islam et al. [18] compared the effects of different heating methods on the proximate composition of freshwater mud eel muscle (*Monopterus cuchia*), and found that the water content of the boiled fish muscle was significantly higher than that of the microwave heating group. Therefore, the reason why the hardness and chewiness of boiled barramundi meat in this study were lower than those treated with MAIH may be due to the fact that boiled meat contained more water and the fish meat texture was softer [18].

### 3.4. Changes in TVBN Content and pH Value of the Barramundi Meat Heated Using Different Methods during Cold Storage

Figure 3 exhibits the changes in the TVBN content and pH value of the barramundi meat that was heated using different methods (i.e., boiling, MAIH at 70 °C, and MAIH at 90 °C) and stored at 4 °C for 60 days. The TVBN content of boiled barramundi meat was initially approximately 4.16 mg/100 g, which increased slowly with the increasing cold storage time and reached 15.6 mg/100 g at the end of cold storage (the 60th day). The TVBN content of the barramundi meat heated using MAIH 70 °C and MAIH 90 °C did not vary obviously during the initial days of cold storage, increased slowly at the later stage of cold storage, and reached 9.80 mg/100 g and 9.57 mg/100 g, respectively, at the end of cold storage (the 60th day). In general, the TVBN value represented an indicator of the freshness of fish meat. Except for elasmobranch fish such as sharks and stingrays, the TVBN content in raw fish or cooked fresh fish muscles is less than 10 mg/100 g [19]. Therefore, this study found that the barramundi meat treated with MAIH was less than 10 mg/100 g during the cold storage period, indicating that the MAIH processing method can maintain better freshness and quality than the boiling method. TVBN is an umbrella term for volatile ammonia, dimethylamine (DMA), trimethylamine (TMA), and biogenic amines [20]. In fresh aquatic products, TVBN is generated by the degradation of protein, amine, and trimethylamine oxide (TMAO), and by the deamination of ATP; its formation pathway is related to the activity of endogenous enzymes and bacterial enzymes [21]. Additionally, rich TMAO contained in aquatic products thermally cracks to produce TMA and DMA when aquatic products are processed at high temperatures [22]. After squid meat was heated at 200 °C for 60 min, its TMAO content decreased sharply, and its DMA content and TMA contents increased by 160% and 370%, respectively [23]. The higher the heating temperature and longer the heating time, the more generated DMA and TMA [24]. Thus, this study inferred that the increase in the TVBN level of boiled barramundi meat might be related to the TMA and DMA generated by the thermal cracking of TMAO. Due to the long heating time in the treatment group of boiling, TMA and DMA were massively generated and gradually released during cold storage, thus increasing the TVBN content. Thus, the MAIH technology (MAIH 70 °C and MAIH 90 °C) slowed down the increase in the TVBN content during cold storage more effectively than the boiling method. These results are similar to those of an earlier finding that the MAIH technology can more significantly slow down the increase in the TVBN content of white shrimps and hard clams, and prolong the storage life of aquatic products more effectively than the conventional boiling method [12,15].

Figure 3B shows the changes in the pH value of barramundi meat that was heated using different methods (i.e., boiling, MAIH at 70 °C, and MAIH at 90 °C) and stored at 4 °C for 60 days. The pH values in the three treatment groups (boiling, MAIH at 70 °C, and MAIH at 90 °C) were initially 6.13, 6.36, and 6.43, respectively, and did not vary significantly (ranging from 6.35 to 6.51) with the increasing cold storage time. Within the same cold storage time, the pH values in the three treatment groups did not significantly differ (*p* > 0.05). These results are similar to earlier findings, which showed that the pH values of MAIH-heated and boiled white shrimps and hard clams do not vary significantly during cold storage, and are not significantly different [12,15].

## 4. Conclusions

This study shows that though no microbial growth was detected in the MAIH-heated and boiled barramundi meat samples, the MAIH technology (MAIH 70 °C and MAIH 90 °C) slowed down the increase in the TVBN content more significantly than the traditional boiling method. Additionally, as the cold storage time increased, lightness (*L**), redness (*a**), and whiteness (*W*) decreased, and yellowness (*b**) and total color difference (Δ*E*) increased in the three treatment groups. However, the MAIH technology (MAIH 70 °C and MAIH 90 °C) slowed down the decrease in lightness, redness, and whiteness more effectively than the traditional boiling method, and maintained a low total color difference (Δ*E*). Moreover, the MAIH technology (MAIH 70 °C and MAIH 90 °C) ensured higher hardness and chewiness of the barramundi meat than the boiling method. Overall, the physicochemical quality of the MAIH-heated fish meat was superior to that of the traditionally boiled fish meat during the period of cold storage. Thus, the MAIH technology is a better and emerging heating method to produce refrigerated ready-to-eat food.

## Figures and Tables

**Figure 1 foods-12-03140-f001:**
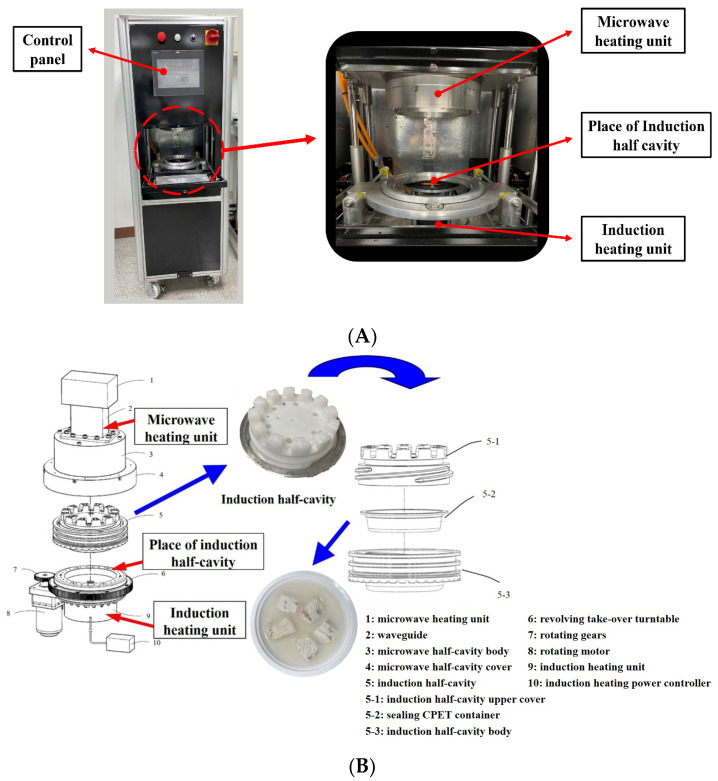
Microwave-assisted induction heating (MAIH) system (Aligo-S, Bottle Top Machinery Co., Ltd., Nantou, Taiwan) (**A**) front view, (**B**) 3D view.

**Figure 2 foods-12-03140-f002:**
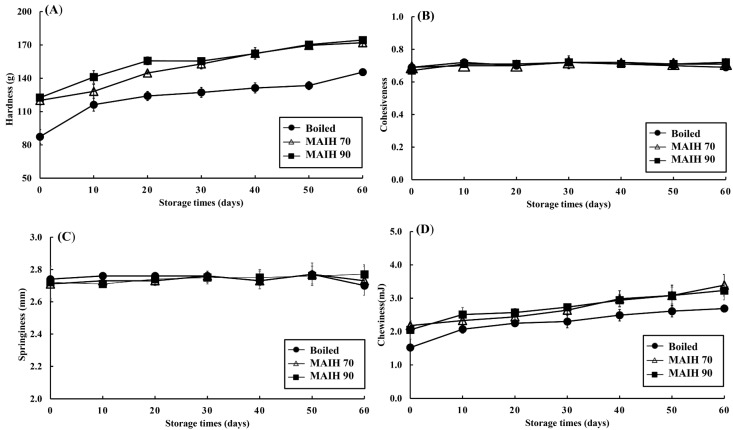
Changes of texture properties of hardness (**A**), cohesiveness (**B**), springiness (**C**), and chewiness (**D**) in barramundi meats using microwave-assisted induction heating (MAIH) system at 90 °C (MAIH 90), 70 °C (MAIH 70), and boiled heating (boiled) stored at 4 °C for 60 days.

**Figure 3 foods-12-03140-f003:**
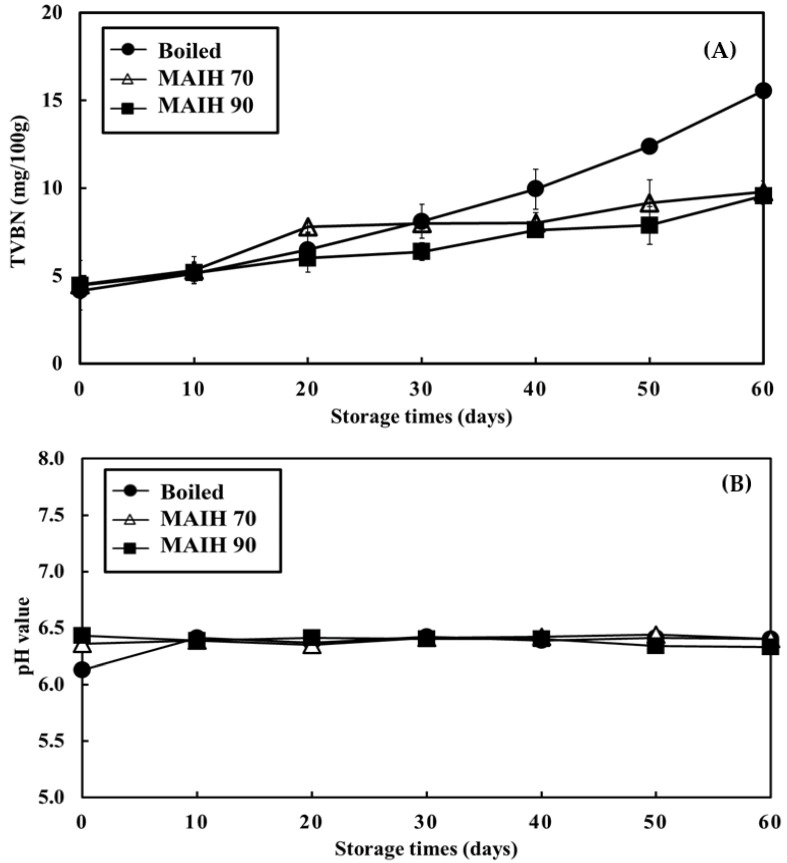
Changes of total volatile basic nitrogen (TVBN) (**A**) and pH value (**B**) in barramundi meats using microwave-assisted induction heating (MAIH) system at 90 °C (MAIH 90), 70 °C (MAIH 70), and boiled heating (boiled) stored at 4 °C for 60 days.

**Table 1 foods-12-03140-t001:** Changes of aerobic plate count (APC), psychotropic bacteria count (PBC), H_2_S-producting bacteria count (HBC), *Vibrio parahaemolyticus*, coliform, and *Escherichia coli* in barramundi meats using MAIH system at 90 °C (MAIH 90) and 70 °C (MAIH 70), and boiled heating (boiled) stored at 4 °C for 60 days.

	Treatments	Storage Time (Days)						
		0	10	20	30	40	50	60
APC	Boiled	<2.0	<2.0	<2.0	<2.0	<2.0	<2.0	<2.0
(log CFU/g)	MAIH 70	<2.0	<2.0	<2.0	<2.0	<2.0	<2.0	<2.0
	MAIH 90	<2.0	<2.0	<2.0	<2.0	<2.0	<2.0	<2.0
PBC	Boiled	<2.0	<2.0	<2.0	<2.0	<2.0	<2.0	<2.0
(log CFU/g)	MAIH 70	<2.0	<2.0	<2.0	<2.0	<2.0	<2.0	<2.0
	MAIH 90	<2.0	<2.0	<2.0	<2.0	<2.0	<2.0	<2.0
HBC	Boiled	<2.0	<2.0	<2.0	<2.0	<2.0	<2.0	<2.0
(log CFU/g)	MAIH 70	<2.0	<2.0	<2.0	<2.0	<2.0	<2.0	<2.0
	MAIH 90	<2.0	<2.0	<2.0	<2.0	<2.0	<2.0	<2.0
*V. parahaemolyticus*	Boiled	<2.0	<2.0	<2.0	<2.0	<2.0	<2.0	<2.0
(log CFU/g)	MAIH 70	<2.0	<2.0	<2.0	<2.0	<2.0	<2.0	<2.0
	MAIH 90	<2.0	<2.0	<2.0	<2.0	<2.0	<2.0	<2.0
Coliform	Boiled	<1.0	<1.0	<1.0	<1.0	<1.0	<1.0	<1.0
(log CFU/g)	MAIH 70	<1.0	<1.0	<1.0	<1.0	<1.0	<1.0	<1.0
	MAIH 90	<1.0	<1.0	<1.0	<1.0	<1.0	<1.0	<1.0
*E. coli*	Boiled	<1.0	<1.0	<1.0	<1.0	<1.0	<1.0	<1.0
(log CFU/g)	MAIH 70	<1.0	<1.0	<1.0	<1.0	<1.0	<1.0	<1.0
	MAIH 90	<1.0	<1.0	<1.0	<1.0	<1.0	<1.0	<1.0

**Table 2 foods-12-03140-t002:** Changes of color (*L**, *a**, *b**, *W,* and Δ*E* value) in barramundi meats using microwave-assisted induction heating (MAIH) system at 90 °C (MAIH 90), 70 °C (MAIH 70), and boiled heating (boiled) stored at 4 °C for 60 days.

Color	Treatments	Storage Time (Days)						
		0	10	20	30	40	50	60
*L**	Boiled	81.74 ± 0.08 aA *^1^	78.64 ± 0.12 bC	78.77 ± 0.26 bC	77.68 ± 0.30 cC	76.37 ± 0.27 eC	76.87 ± 0.08 dB	75.81 ± 0.18 fC
	MAIH 70	81.88 ± 0.22 aA	80.89 ± 0.04 bA	79.93 ± 0.06 cA	79.45 ± 0.07 dA	79.46 ± 0.47 dA	78.76 ± 0.37 eA	79.18 ± 0.21 deA
	MAIH 90	81.19 ± 0.77 aA	80.61 ± 0.16 aB	79.36 ± 0.17 bB	78.60 ± 0.38 bcB	78.54 ± 0.58 bcB	78.14 ± 0.99 cA	78.31 ± 0.63 dB
*a**	Boiled	1.53 ± 0.15 aB	1.20 ± 0.15 bB	1.07 ± 0.03 bB	0.69 ± 0.03 cB	0.64 ± 0.06 cB	0.58 ± 0.04 cB	0.57 ± 0.01 cB
	MAIH 70	2.00 ± 0.11 aA	1.53 ± 0.05 bA	1.36 ± 0.21 bcA	1.16 ± 0.09 cdA	1.08 ± 0.05 dA	0.94 ± 0.24 dA	0.95 ± 0.11 dA
	MAIH 90	1.60 ± 0.03 aB	1.49 ± 0.04 aA	1.27 ± 0.03 bAB	1.12 ± 0.02 cA	1.08 ± 0.08 cdA	1.06 ± 0.11 cdA	0.93 ± 0.16 dA
*b**	Boiled	10.61 ± 0.13 eB	11.62 ± 0.16 dA	11.39 ± 0.31 dB	11.85 ± 0.58 cdB	12.23 ± 0.19 cB	13.45 ± 0.04 bA	14.18 ± 0.03 aA
	MAIH 70	11.02 ± 0.07 fA	11.58 ± 0.17 eA	12.12 ± 0.07 dA	13.15 ± 0.11 cA	13.27 ± 0.30 cA	13.59 ± 0.16 bA	14.32 ± 0.21 aA
	MAIH 90	11.24 ± 0.14 dA	11.47 ± 0.01 dA	12.14 ± 0.08 cA	13.26 ± 0.16 bA	13.57 ± 0.14 bA	13.81 ± 0.73 bA	14.43 ± 0.45 aA
*W*	Boiled	78.53 ± 0.36 aA	74.18 ± 0.05 eB	75.85 ± 0.27 cB	74.58 ± 0.19 dC	74.16 ± 0.18 eB	74.46 ± 0.13 bC	74.37 ± 0.14 deB
	MAIH 70	78.54 ± 0.07 aA	78.62 ± 0.47 aA	76.72 ± 0.52 bA	75.70 ± 0.15 cA	75.72 ± 0.57 cA	75.25 ± 0.07 cA	75.36 ± 0.31 cA
	MAIH 90	78.79 ± 0.07 aA	77.99 ± 0.63 bA	76.48 ± 0.31 cAB	75.25 ± 0.08 deB	75.75 ± 0.12 dA	75.30 ± 0.05 fB	75.05 ± 0.25 eA
Δ*E*	Boiled	-	3.28 ± 0.18 dA	3.11 ± 0.17 dA	4.33 ± 0.45 cA	5.68 ± 0.19 bA	5.71 ± 0.04 bA	6.98 ± 0.14 aA
	MAIH 70	-	1.24 ± 0.02 dB	2.33 ± 0.04 cB	3.34 ± 0.01 bB	3.45 ± 0.15 bB	4.18 ± 0.44 aB	4.40 ± 0.06 aB
	MAIH 90	-	0.64 ± 0.15 cC	2.07 ± 0.12 bC	3.33 ± 0.20 aB	3.57 ± 0.53 aB	4.04 ± 1.20 aB	4.36 ± 0.73 aB

*,^1^ All data were the means ± standard deviation of three replicates. A–C: different letters in the same column indicate significant differences (*p* < 0.05); a–f: different letters in the same row indicate significant differences (*p* < 0.05).

## Data Availability

The data presented in this study are available on request from the corresponding author. The data are not publicly available due to privacy and ethical reasons.

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
