# Peer review of "Physicochemical Quality Retention during Cold Storage of Prepackaged Barramundi Meat Processed with a New Microwave-Assisted Induction Heating Technology"

_foods, 2023, doi:10.3390/foods12163140_

Round 1

Reviewer 1 Report

The manuscript titled “Physicochemical quality retention during cold storage of pre-packaged barramundi meat processed with a new micro-wave-assisted induction heating technology” describes the new processing technology of micro-wave-assisted induction heating at 90℃ for 110 seconds and 70℃ for 130 seconds of barramundi fish meat and stored at 4°C. The authors analyzed a variety of physico-chemical parameters of the stored fish at 10 days interval including microbial count, color attributes (L*, a*, b*, W, and ΔE), texture, total volatile basic nitrogen (TVBN) content, and pH value. In all types of stored fish meat, no microbial growth was observed, and the microwave-assisted induction heating (MAIH) technology was found to significantly slow down the quality deterioration of barramundi meat. The MAIH method also maintained good color and texture of the fish during cold storage. Based on these findings, the authors concluded that MAIH is a superior and emerging heating method for producing refrigerated ready-to-eat food.

The research holds great interest and impact in terms of processing fish muscles and their storage. The manuscript is well-designed and written with a sound structure. However, there is some minor spelling and grammatical errors that require thorough checking.

Specific comments:

1.     Abstract: Line 15; make it correct 70°℃.

2.     Abstract: Line 27; What do the authors mean by good colour. Make it clear please.

3.     Scientificnames should be Italic. E.g., Lates calcarifer in Line 103; Vibrio parahaemolyticus in Line 135; Escherichia coli in Line 135). Check the whole manuscript and make correction.

4.     3.4. Changes in TVBN content (Line: 296): The authors should state the references value of TVBN content of similar product.

5.     The authors may find some information to enrich the discussion in the following paper:

Nutritional characterization of freshwater mud eel (Monopterus cuchia) muscle cooked by different thermal processes

-

Author Response

Response to Reviewer 1 Comments:

  1. Abstract: Line 15; make it correct 70°.

Response 1Revised as suggested. The "70°℃" was revised to "70℃". Please see L. 15.

  1. Abstract: Line 27; What do the authors mean by good colour. Make it clear please.

Response 2Thanks for the review’s comment. The "good colour" was revised to "better colour" to suitable for the meaning for reader. Please see L. 27.

  1. Scientificnames should be Italic. E.g., Lates calcarifer in Line 103; Vibrio parahaemolyticus in Line 135; Escherichia coli in Line 135). Check the whole manuscript and make correction.

Response 3Revised as suggested. The " Vibrio parahaemolyticus " and " Escherichia coli " were revised to " Vibrio parahaemolyticus " and " Escherichia coli " as Italic in throughout text. Please see L. 128, 129, 141-142, 192-193, and 194-195.

  1. 3.4. Changes in TVBN content (Line: 296): The authors should state the references value of TVBN content of similar product.

Response 4Thanks for the review’s comment. This is achieved by adding some statements, as the following " In general, the TVBN value represented an indicator of the freshness of fish meat. Except for elasmobranch fish such as sharks and stingrays, the TVBN content in raw fish or cooked fresh fish muscles was less than 10 mg/100 g [19]. Therefore, this study found that the barramundi meat treated with MAIH was less than 10 mg/100 g during the cold storage period, indicating that the MAIH processing method can maintain better freshness and quality than the boiling method.". Please see L. 307-312.

  1. The authors may find some information to enrich the discussion in the following paper: Nutritional characterization of freshwater mud eel (Monopterus cuchia) muscle cooked by different thermal processes

Response 5Thanks for the review’s comment. This reference of the Islam et al. [18] was added in the references. And the discussion compared with this paper was added into the text, as following " Islam et al. [18] compared the effects of different heating methods on the proximate composition of freshwater mud eel muscle (Monopterus cuchia), and found that the water content of the boiled fish muscle was significantly higher than that of the microwave heating group. Therefore, the reason why the hardness and chewiness of boiled barramundi meat in this study are lower than those treated with MAIH may be due to the fact that boiled meat contains more water and the fish meat texture is softer [18].". Please see L. 286-292.

Reviewer 2 Report

There are some errors in language and grammar. A native English speaker can easily correct these difficulties, some of which are noted in the comments. The missing information on procedures prevent determining if they are appropriate and do not allow duplication of the experiments by other scientists was missing.

Line(s)      Comment

104           The term “trash ice” is not understood.

105           The anatomical location (dorsal?) and/or muscle name for the samples should be given.

106           The vapor permeability or thickness and the source of the CPET tray should be given.

108           The vapor permeability or thickness and the source of the PET film should be given.

122           The material of the aseptic bag and vapor permeability or thickness should be given.

127-128   Figure 2 shows sampling every 20 days so “They were sampled at 10 day intervals to analyze”

153           The aperture opening size, optical geometry, illumination type, and observer angle must be given.

186           The number of replications of the experiment should be given. A one-way ANOVA is not appropriate for analyzing the data in a completely randomized design with n replications, 3 treatments and 7 storage times unless there were no interaction effects (replication x treatment, replication x storage time, or treatment x storage time). If replication is not significant, then a 2-way ANOVA (3 treatments and 7 storage times) must be used.

153           “data”

229-230   Use of “significantly” and the probability in the same sentence here and in subsequent sentences is redundant.

270           “significantly slowed the decline”

272           “clam meat was higher than”

285-286   “significantly slowed the decline”

288           “clam meat was higher”

A native English speaker can easily correct the minor language and grammar errors and use of present rather than past verb tense when previous work is discussed.

Author Response

Response to Reviewer 2 Comments:

  1. 104    The term “trash ice” is not understood.

Response 1Thanks for the review’s comment. The "trash ice" was revised to "crushed ice". Please see L. 94.

  1. 105   The anatomical location (dorsal?) and/or muscle name for the samples should be given.

Response 2Thanks for the review’s comment. The "back" was revised to "dorsal part". Please see L. 95.

  1. 106   The vapor permeability or thickness and the source of the CPET tray should be given.

Response 3Thanks for the review’s comment. We added the " and 0.25 cm in thickness " and " All CPET boxes and PET films were supported by Bottle Top. Machinery Co., LTD., Taiwan. " in text. Please see L. 98 and 100-101.

  1. 108   The vapor permeability or thickness and the source of the PET film should be given.

Response 4Thanks for the review’s comment. We added the " (80 µm in thickness) " and " All CPET boxes and PET films were supported by Bottle Top. Machinery Co., LTD., Taiwan. " in text. Please see L. 100-101.

  1. 122   The material of the aseptic bag and vapor permeability or thickness should be given.

Response 5Thanks for the review’s comment. We added the " (polyethylene, 63.5 µm in thickness) " in text. Please see L. 115.

  1. 127-128   Figure 2 shows sampling every 20 days so “They were sampled at 10 day intervals to analyze”

Response 6Thanks for the review’s comment. It is corrected for sampling by every 10 days, not for 20 days. Please see the scale of X-axis in Figure 2.

  1. 153    The aperture opening size, optical geometry, illumination type, and observer angle must be given.

Response 7Thanks for the review’s comment. We added the " with 30 mm aperture opening size, D/8 optical geometry, 100 viewing angle, and illumination of D65" in text. Please see L. 148-149.

  1. 186   The number of replications of the experiment should be given. A one-way ANOVA is not appropriate for analyzing the data in a completely randomized design with n replications, 3 treatments and 7 storage times unless there were no interaction effects (replication x treatment, replication x storage time, or treatment x storage time). If replication is not significant, then a 2-way ANOVA (3 treatments and 7 storage times) must be used.

Response 8→Thanks for the review’s comment. We added the " all values were derived as the average ± standard deviation of three individual containers or bags and " in text. Please see L. 181-182. In addition, there were no interaction effects in this study. So, one-way ANOVA is correct.

  1. 153    “data”

Response 9→Revised as suggested. The "datam" was revised to "data". Please see the footnote of Table 2 in L. 248.

  1. 229-230   Use of “significantly” and the probability in the same sentence here and in subsequent sentences is redundant.

Response 10→Revised as suggested. The "significantly" was deleted and "(p<0.05)" was reserved. Please see L. 225, 228 and 230.

  1. 270   “significantly slowed the decline”

Response 11→Revised as suggested. Please see L. 265.

  1. 272           “clam meat was higher than”

Response 12→Revised as suggested. Please see L. 267.

  1. 285-286   “significantly slowed the decline”

Response 13→Revised as suggested. Please see L. 280.

  1. 288    “clam meat was higher”

Response 14→Revised as suggested. Please see L. 283.

  1. A native English speaker can easily correct the minor language and grammar errors and use of present rather than past verb tense when previous work is discussed.

Response 15Thanks for the review’s comment. The minor language and grammar errors and use of present rather than past verb tense were revised in the throughout text. Please see L. 65, 66, 68, 73, 74, 76, 80, 234, and 359.

Reviewer 3 Report

Dear Authors,

The topic of the manuscript entitled ” Physicochemical quality retention during cold storage of pre-

packaged barramundi meat processed with a new micro-wave-assisted induction heating technology” is interesting but needs some changes.

Major comments:

Some of my comments are:

Introduction – Information about the operation principle of a microwave oven is unnecessary. The goal is not to describe the operation principle but rather the mechanism of how waves affect changes in the meat. Please remove Figure 1.

In method section – please add information about repetitions.

Author Response

Response to Reviewer 3 Comments:

  1. Introduction – Information about the operation principle of a microwave oven is unnecessary. The goal is not to describe the operation principle but rather the mechanism of how waves affect changes in the meat. Please remove Figure 1.

Response 1Thanks for the review’s comment. The operation principle of a microwave oven in Introduction was deleted. Please see L. 38 for track changes and L. 56 for track changes. In addition, due to MAIH equipment as a new heating technology, the Figure 1 is very important for reader to realize the principle of processing. Therefore, Figure 1 was reserved in this manuscript.

  1. In method section – please add information about repetitions.

Response 2Thanks for the review’s comment. We added the " all values were derived as the average ± standard deviation of three individual containers or bags and " in 2.8. Statistical analysis. Please see L. 181-182.

Round 2

Reviewer 2 Report

The revisions are appreciated, but without stating the number of replications of the experiment or if there were interactions between the microwave treatments and storage times, the manuscript is not acceptable for publication because the inference space is restricted to only this study and is not applicable to any other similar conditions.

The language is improved.
